# Changes in Access to Health Services during the COVID-19 Pandemic: A Scoping Review

**DOI:** 10.3390/ijerph19031749

**Published:** 2022-02-03

**Authors:** Georgina Pujolar, Aida Oliver-Anglès, Ingrid Vargas, María-Luisa Vázquez

**Affiliations:** Health Policy Research Unit (SEPPS), Consorci de Salut i Social de Catalunya, 08022 Barcelona, Spain; aoliver@consorci.org (A.O.-A.); ivargas@consorci.org (I.V.); mlvazquez@consorci.org (M.-L.V.)

**Keywords:** COVID-19, health services’ accessibility, delivery of health care, health care inequalities

## Abstract

The COVID-19 pandemic and the measures adopted are having a profound impact on a major goal of public healthcare systems: universal access to health services. The objective is to synthesize the available knowledge on access to health care for non-COVID-19 conditions and to identify knowledge gaps. A scoping review was conducted searching different databases (Medline, Google Scholar, etc.) for original articles published between December 2019 and September 2021. A total of 53 articles were selected and analyzed using the Aday and Andersen framework as a guide. Of these, 37 analyzed changes in levels of use of health services, 15 focused on the influencing factors and barriers to access, and 1 studied both aspects. Most focused on specific diseases and the early stages of the pandemic, based on a review of records. Analyses of the impact on primary care services’ use, unmet needs or inequalities in access were scarce. A generalized reduction in the use of health services was described. The most frequent access barrier described for non-COVID-19 conditions related to the services was a lack of resources, while barriers related to the population were predisposing (fear of contagion, stigma, or anticipating barriers) and enabling characteristics (worse socioeconomic status and an increase in technological barriers). In conclusion, our results show a general reduction in services’ use in the early stages of the pandemic, as well as new barriers to access and the exacerbation of existing ones. In view of these results, more studies are required on the subsequent stages of the pandemic, to shed more light on the factors that have influenced access and the pandemic’s impact on equity of access.

## 1. Introduction

The pandemic due to coronavirus SARS-CoV-2 (COVID-19), a novel virus initially reported in December 2019 [1], was declared by the World Health Organization (WHO) on 11 March 2020. It has had an adverse effect worldwide on many different spheres of society, including the economy and public health. The current COVID-19 pandemic and the diverse strategies that have been adopted to tackle it are forcing changes in access to health services for other conditions, potentially producing an impact on the health of the population above and beyond that caused by COVID-19 itself [2,3,4,5,6].

In this regard, some strategies taken to combat soaring COVID-19 infection rates may have negatively affected access to health services for other conditions. Firstly, at the health services level, one of the most influential measures was the classification of services as essential or non-essential, following WHO guidelines, which allowed resources to be redirected to the pandemic response. However, this has also caused cancellations or delays in elective and non-urgent procedures [3,5,6,7], despite many countries implementing strategies to mitigate the impact of these disruptions (e.g., online healthcare visits) [4,5], Another significant measure was social distancing to reduce interaction between people, including nationwide partial or complete lockdowns, schools and non-essential business closures, and instructions to stay at home, which in some cases has erected a barrier in terms of mobility to entry the health services [8].

Another indirect effect of the pandemic, the economic crisis stemming from the substantial curtailment of economic activity, and the ensuing rise in unemployment and loss of household income, have aggravated associated access barriers (loss of health coverage, difficulties in making copayments or obtaining transport to services), thereby accentuating existing inequalities in access, as studies on previous economic crises have shown [9]. Although it is necessary to evaluate which population groups have been particularly affected in terms of access to care and how the determining factors interact with each other, there are some signs—including early evidence and experiences from previous crises—to indicate that vulnerable population groups (populations with low socioeconomic status, the elderly, chronic patients or those with severe conditions, migrants from low-income countries) suffer a greater impact [9,10,11,12,13,14].

In short, as in other epidemics and previous outbreaks, the health repercussions of the current pandemic are not confined solely to COVID-19 infection and mortality. They also include indirect negative effects on healthcare access and on the quality of curative and preventive care provided for other conditions, and the exacerbation of difficulties and barriers related to socioeconomic factors [15,16,17]. The scientific evidence accumulated from previous experiences, such as severe acute respiratory syndrome or SARS (2002–2003), Middle East respiratory syndrome or MERS (2012), Ebola (2014–2016; 2018–present), and the Zika virus (2015–2016) [18,19,20], shows a decrease in the utilization of health services (e.g. outpatient care, hospital admissions, elective surgeries [20,21,22,23]) that is attributed to changes made to the health services in response to public health emergencies, as well as to fear of contagion among the population [21,24]. This may in turn have had an impact on increasing the burden of disease and mortality in the months following an epidemic outbreak [6,20,25,26,27].

While a plethora of scientific papers have been published on COVID-19 since the start of the pandemic, studies on its impact on access to health services have not been so plentiful. A few literature reviews have been found that summarize changes in health services due to the pandemic, focusing mainly on the adoption of telemedicine [28,29,30,31,32] and the impact of the pandemic on different aspects, among others, the use of certain specific services, such as maternal and child health care [33], child vaccination [34], or chronic diseases [35,36], in the initial stages of the pandemic. Although we are still at the pandemic management stage, a synthesis of the scientific evidence accumulated to date on the impact on access to health care in different contexts, in addition to detailed monitoring of the performance of services, may help decision makers to make healthcare systems more resilient in current and future emergencies and protect public health and access to health care.

Access to care involves many highly interdependent factors and stakeholders at play [37]. This study takes as its framework of reference one of the most frequently used models for the analysis of access to health services [38], that of Aday and Andersen [39]. This model distinguishes between realized access (effective utilization of the services) and potential access (determinants of access), differentiating between factors related to the services and to the population. The realized access analysis takes into account the type, place, motive of the visit (preventive or curative), and care outcomes, while potential access analysis takes into account the characteristics of the services (availability of resources and organization) and those of the population (predisposing factors: sociodemographic factors, beliefs, attitudes, and knowledge of the health system; enabling factors: income and type of insurance; health needs). Health policies, programs, or interventions can in turn affect access barriers related to the health services or changing (mutable) characteristics of the population [39,40]. Aday and Andersen’s framework is more comprehensive and exhaustive than other analytical models [41,42,43], which focus either on the entry to health services or on the characteristics of services and how they adjust to the characteristics of the population. Thus, it offers an appropriate approach for identifying existing knowledge gaps in the literature on access and to analyzing different types of barriers and factors that influence the use of health services.

The aim of this article is to synthesize the knowledge accrued from the onset of the pandemic in March 2020 through to September 2021 on the impact of the COVID-19 pandemic on access to health services (including primary care, outpatient secondary care, and inpatient care) for non-COVID-related conditions, and to identify knowledge gaps on these subjects.

## 2. Materials and Methods

A scoping review of the scientific literature [44] was carried out, following the PRISMA guide [45], to identify original articles on the impact of the COVID-19 pandemic on access to health services for non-COVID-related conditions.

In our bibliographic search, several digital databases were consulted to minimize the risk of overlooking any relevant studies: Medline, Google Scholar, SCiELO, and Web of Science. The search was performed over two separate periods: 22 January 2021–31 March 2021 and 22 September 2021–10 October 2021. In the Medline database, using a thesaurus, MeSH terms were employed for: (a) COVID-19: “Coronavirus Infections”, “Coronavirus”, “COVID-19”, “SARS-CoV-2”; (b) access to health services: “Health services availability”, “Health services needs and demand”, “Healthcare disparities”, “Needs assessment, healthcare”, “Health policy”, “Equipment and Supplies Utilization”, “Facilities and Services Utilization” (see Appendix A for more detail). In the other databases free text terms were used: (a) COVID-19: “coronavirus disease”, “COVID-19”; (b) access to health services: “health services accessibility”, “accessibility”, “accessing”, “access”, “utilization”, “delivery of health care”, “healthcare services”, among others. Word groups were combined using Boolean operators “AND” and “OR” in order to identify the literature in the databases consulted and select those studies referring to the impact of COVID-19 on access to health services. The search was complemented with a manual review of references cited within the selected articles.

We selected original articles published in Spanish or English from December 2019 to September 2021—with no filters for geographical area or motive for access (apart from being unrelated to COVID-19)—that used qualitative and/or quantitative methods and analyzed or described changes in access to health services in the context of the COVID-19 pandemic. The initial selection of studies to review was performed through title and abstract screening. Where there was any doubt about whether to include a study, this was discussed with another researcher in the team. 

Following Aday and Andersen’s framework [39,40], the selected studies were classified into two groups: those that analyzed utilization of services (realized access) and those that explored factors that influence access (potential access). Any studies on realized access that did not use medical records, administrative/institutional databases, or patient surveys as their data source were excluded. A data extraction protocol was produced to include information related to methodological aspects (methods, period of analysis, study area, population, sample, type of health service) and study results (according to the variables or dimensions of analysis). This information was extracted from the articles and presented in tables, grouped according to type of access, and ordered by type of health service. The results were summarized according to the analytical framework, which was also used to identify any gaps in knowledge related to the aspects studied.

## 3. Results

From the search results, 242 articles were identified for title and abstract screening, and 95 for full-text review. A total of 53 articles met the inclusion criteria for analysis (Figure 1).

Of the 53 articles selected, 37 analyzed changes in realized access applying quantitative methods [46,47,48,49,50,51,52,53,54,55,56,57,58,59,60,61,62,63,64,65,66,67,68,69,70,71,72,73,74,75,76,77,78,79,80,81,82], through the analysis of medical records (Table 1), 8 analyzed changes in potential access [83,84,85,86,87,88,89,90] via surveys of different population groups, except for one that did the same via analysis of medical records [90] (Table 2), 7 used qualitative methods to analyze the impact on potential access [91,92,93,94,95,96,97] (Table 3), and, finally, 1 study analyzed changes in both realized and potential access [98] using mixed methods (analysis of medical records and semi-structured interviews). Only six of the studies that explored changes in potential access included professionals from the health centers analyzed as a study population, [87,89,92,93,97,98], while the rest focused on patients or the general population.

With regard to the type of service, of the studies on realized access, 5 focused on health services in general [46,47,48,49,50]; 12 on emergencies [51,52,53,54,55,56,57,58,59,60,61,62] (of which 5 were related to pediatric emergencies [56,57,58,60,61]); 15 on secondary care (SC) (outpatient visits, hospital admissions, etc., for nephrology [63], oncology [64,65], pediatrics [66], psychiatry [67,68], rehabilitation [69], respiratory diseases [70], sexual and reproductive health (SRH) [71,73,74,75], and traumatology [78,79]); two on primary care (PC) [80,81]; one on preventive services [82]. Of the studies on potential access, 1 focused on health services in general [91], 13 on SC (endocrinology [83], rare diseases [86], pediatrics [84], psychiatry [85], SRH [87,88,93,94,95], tuberculosis [89,96] and urology [90]); and 2 on PC [92,97]. The study that analyzed changes in both realized and potential access focused on SC relating to SRH [98].

In terms of geographical area, 19 studies were conducted in European countries [47,51,53,55,57,62,64,67,68,69,70,75,79,83,84,88,90,92,95], 9 in North American countries [49,58,59,77,78,80,82,85,86], 9 in Sub-Saharan Africa [46,48,71,72,76,87,93,94,98], 3 in Latin America [60,91,96], 5 in the East Asia–Pacific region [50,56,61,66,81], 3 in South Asia [63,65,74], 2 in the North Africa–Middle East region [52], and, lastly, 3 studies covered various regions [54,89,97].

With regard to the period of analysis, 38 of the selected studies were conducted over the first months of the pandemic (February to June 2020) [46,47,48,49,50,51,52,53,54,55,57,58,59,60,62,65,66,67,68,69,70,72,73,74,75,76,77,78,79,80,82,84,86,88,90,92,93,98]; 4 during the second stage of the pandemic (June to September 2020) [85,89,91,96]; 1 at a later stage (September to November 2020) [94]; finally, 8 analyzed the whole period (February to December 2020) [56,61,63,64,71,81,83,87]. No studies were found that analyzed periods after December 2020. All studies that analyzed realized access presented their results in comparison with a previous period: 32 of the first-stage studies compared the changes to pre-pandemic periods of reference [46,47,48,49,50,51,52,53,54,55,57,58,59,60,62,65,66,67,68,69,70,72,73,74,75,76,77,78,79,80,82,98], as did 6 of those that analyzed the whole period [56,61,63,64,71,83]. One study did not specify the exact period of 2020 analyzed [95].

Below is a summary of results found regarding changes in realized access and potential access, following the Aday and Andersen theoretical framework [39,40].

### 3.1. Changes in the Utilization of Health Services and Influencing Factors

Of the 38 studies that analyzed changes in realized access [46,47,48,49,50,51,52,53,54,55,56,57,58,59,60,61,62,63,64,65,66,67,68,69,70,71,72,73,74,75,76,77,78,79,80,81,82,98], 33 indicated a statistically significant descent in the use of services and only one reported an increase [78], and 4 descriptive studies also found a reduction in the use of services [47,61,63,65] (Table 1). After the first few months of the pandemic, some studies described an increase in the utilization, without reaching levels previous to the COVID-19 pandemic [48,61,63,71,72,82]. However, among the studies that extended their analysis to the end of 2020, there are reports of new drops in the utilization of services, coinciding with the onset of new waves of COVID-19 [61,63,71].

By type of service analyzed, studies focusing on the health services in general [46,47,49,50] described an overall drop in use, which varied in terms of magnitude. Among those that analyzed changes in emergency care [51,52,53,54,55,56,57,58,59,60,61,62,66,77], it was reported that, although the volume of consultations fell, there was an increase in the number of cases with additional complications [53,58,66,74,77] and cases requiring admission to hospital [51,55,57,59,60,67,70]. Likewise, the studies on changes in SC services also reported a reduction in the use of services, while in some cases detecting an increase in the proportion of visits or hospital admissions in concrete healthcare areas (e.g., severe mental health cases [47] or trauma injuries [56,58,78]). With respect to PC, both studies described a drop in the number of in-person visits and an increase in remote care consultations (via various mechanisms, such as consultation by telephone or videocall) [66,80]. Lastly, the study that analyzed changes in the use of preventive services [82], along with others that also assessed procedures classified as elective [49,53,63,64,77], reported a drop in use but with less pronounced changes in urgent cases [57,62,82].

In the studies that analyzed influencing factors in the use of services in the context of the pandemic, the probability of lower utilization levels was associated with different factors. With regard to predisposing characteristics of the population, women [56,79] and ethnic minorities [49,74] were less likely to access health services, with inconsistent results regarding the elderly [52,79]. As for individual enabling characteristics, the likelihood of a lower use of services was associated with people with a low income and limited healthcare coverage [49,74], and for those enabling characteristics related to the type of area, a higher incidence of COVID-19 [50]. In terms of need, the probability of using health services was lower among patients who lacked a previous diagnosis [82], had less severe conditions (without complications or adverse outcomes) [67], and did not require hospitalization [52,57,67].

### 3.2. Potential Access: Barriers Related to Characteristics of the Services and Population

The 16 studies that analyzed potential access [83,84,85,86,87,88,89,90,91,92,93,94,95,96,97,98] described changes in the factors that influenced access before the COVID-19 pandemic and the emergence of new barriers, in terms of both the characteristics of the services and those of the population (Table 2 and Table 3).

Twelve studies described changes related to characteristics of the services [83,85,86,87,88,89,90,92,93,95,97,98]. The most significant of these was a reported decrease in available resources, both in terms of materials and medical supplies [83,86,89,93,97], and of staff to care for non-COVID patients [83,85,88,89,92,95], which in certain cases forced some health centers to close [93,97]. Two studies pointed out increases in waiting times [90,98] and one a rise in the cost of services [97].

Fourteen studies identified barriers related to population characteristics [83,84,85,86,87,88,89,91,93,94,95,96,97,98]. Among the predisposing factors identified, fear of contagion was one of the main reasons for not going to the health services [83,84,85,86,88,89,94,95,96,97,98], as well as the stigma that receiving a COVID-19 diagnosis would create [89,94,97]. Other factors found included people misinterpreting government recommendations to avoid going to the health services [84], perceiving that the services were of poor quality [94], or believing they would have difficulties in gaining access [85,86,91,93] during the pandemic. With respect to enabling factors, authors highlighted the worsening socioeconomic situation of the population [87,89,91,93,94,96,97], a lack of support networks [93,94], and an increase in technological barriers [85,88] as some of the main factors that hindered access to the health services. Lastly, they also underlined tendencies to play down the risk of health complications and the perceived need for medical attention [94,98] as barriers that had the effect of reducing the use of services and delaying the decision to seek care during the pandemic.

## 4. Discussion

The impact of the COVID-19 pandemic has been felt worldwide in many different spheres of society, but especially in access to health services for unrelated conditions. There is now a pressing need to evaluate the changes that have arisen in this regard, and their implications for equity of access and the resilience of national health systems to future pandemics. This is, to the best of our knowledge, the first scoping review to offer a general overview of the subject, taking in the current evidence and highlighting the areas that will require further research in future studies.

Most of the studies included in the analysis describe a lower level of health services’ utilization and changes in potential access, as preexisting barriers have intensified and new ones have arisen. However, while investigations into the impact of the COVID-19 pandemic are still ongoing, the results of this review show that the studies conducted to date are limited in terms of scope and methodology, and that they are mainly centered on analyzing the impact on the use of services for specific diseases or population groups during the first stages of the pandemic, with a particular focus on secondary care.

Studies on the use of health services in general are very scarce, as are those on access to primary care, which in many countries has been the most overwhelmed care level due to having to take on more pandemic-related care duties (vaccination, case tracking, etc.). Likewise, there is a considerable lack of evidence so far on how changes have taken place over the course of the different waves of COVID-19, and according to geographical context. Although some studies with longer timeframes—to the end of 2020—have already described new drops in health services’ activity following brief periods of recovery [61,63,71], further evidence is required to confirm this trend. Moreover, we have yet to look into how the pandemic has affected unmet care needs, as some studies showed that one of the most commonly reported impact was that patients delayed seeking care due to factors such as fear of contagion, disinformation, etc. [83,84,85,86,88,89,94,95,97,98].

It is important to bear in mind that access to care involves multiple interdependent factors and numerous actors. However, the results of this review show that, so far, there are almost no published studies with a wide scope using mixed methods, and that the qualitative studies available to date are still limited in both number and perspective. In regard to the latter, few studies include, in addition to that of users, other viewpoints such as that of the health professionals or managers involved in the process of taking measures or adopting practices that influenced access to care. Furthermore, the population groups selected for study were generally sufferers of a specific condition or of vulnerable status. The number of qualitative studies is probably limited due to the complexity of their development in the pandemic context, in terms of time, resources, and restrictions imposed by the mitigation measures. Approaches that combine multiple sources of evidence and different perspectives are needed to shed more light on the factors and actors that have influenced access.

With regard to our main findings on the reduction in the use of services, this may be related to health systems prioritizing their response to the public health emergency, which differed according to context [4,25,99,100]. Initial measures were generally centered on containing the spread of COVID-19 and providing the health services with the resources needed to meet the soaring demand for medical attention, which led to the classification of some services and procedures as non-essential, and a consequent reduction in the resources allocated to cover those health needs [3,4,7,8]. There is a lack of evidence on how the measures were modified and adapted according to context as the pandemic progressed, and the patterns of utilization that they generated, although some studies have already revealed changes in use with recoveries and relapses, in the more advanced stages of the pandemic [61,63,71].

Going into more detail, the reductions in access to services described appear to have brought with them an increase in medical complications and emergencies, especially in elective procedures [48,49,52,53,58,59,67,71,74,77,101], and/or care delays (time passed between onset of symptoms and intervention) [53,98], although it is generally not specified whether the delays were due to patients putting off seeking medical attention or rather to an increase in barriers to access the services. Some studies also observed higher mortality rates [52,70,74,76] and burden of disease [62]. However, while emergency care was one of the greatest causes for concern [51,52,53,54,55,56,57,58,59,60,61,62], according to our results the downturns recorded for this type of service appear to be less pronounced or of lower impact than those reported in other fields.

In this regard, some studies highlight the difficulties involved in maintaining normal levels of activity in certain services, even in some classed as essential, such as maternal health, oncology, or mental health [71,76,85,89,93,94,97,98]. The impact appears to have been felt worldwide but especially in middle- and low-income countries, a point that has also been stressed in various opinion articles [101,102,103,104,105,106,107]. Differences between health systems, and between geographical contexts, may both have acted as determining factors in the changes seen. Sexual and reproductive health services analyzed in African and South Asian countries, for example, have seen a significant downturn in access, not only to maternal and child health services but also other non-essential areas of health care (family planning, sexually transmitted diseases, etc.) [71,74,76,93,98], despite the warnings given in various reports and opinion articles based on previous epidemic experience [2,26,101,108] of the risk this poses in terms of burden of disease and mortality [2,25,26,103,104,105,106,109,110,111,112,113,114]. Another example can be found in the lack of care for patients at risk of a cancer diagnosis due to delays in screening and diagnostic programs caused by certain procedures being classified as elective [64,65,82], which could also lead to an increase in the burden of disease and mortality due to the late detection of new cancer cases, as several studies pointed out [35,115,116,117,118]. As regards mental health, various studies mention anxiety and other disorders related to fear of contagion and the uncertainty that the pandemic generated in its first few months [35,55,77,83,84,85,86,88,91,94,95,96], as well as difficulties in access to mental health services [68,85,97] and an increase in acute cases of severe disorders [67]. These issues may have caused the burden of disease to increase, which could influence service utilization patterns in the years to follow, an aspect that was observed previously with the SARS epidemic in 2003 [119] and that should be taken into consideration in future studies on the impact of COVID-19.

The few studies that analyze the influencing factors on the lower utilization of services mainly highlight a greater downturn in use among low-income users, those with limited healthcare coverage, and ethnic minorities [49,74], as well as the female population [56,79], all of which signals greater inequalities with regard to more vulnerable populations.

As regards the changes in potential access detected in this review, the results indicate both the exacerbation of existing barriers, related above all to structural difficulties and situations of vulnerability, and the creation of new ones. In terms of existing barriers accentuated by the pandemic, some studies reported a shortage of resources in the services to meet all the incoming healthcare requests [83,85,86,88,89,92,93,95,97,98], which varied according to service and geographical context. One of the most serious problems was the lack of or alterations to the distribution of materials and medical supplies in low-income countries [83,87,89,93,94,97], an aspect corroborated in other publications, both reports and opinion articles [4,34,104,120,121,122]. The lack of materials and medical supplies may have contributed to increasing negative perceptions of the quality of the health services, especially in disadvantaged settings or situations with structural difficulties, another barrier to access found by some studies [93,94,97]. Lastly, several studies in this review [83,85,87,89,91,93,97] and some reports [12,123] focusing on vulnerable population groups (such as migrants or refugees, sex workers, people at a severe socioeconomic disadvantage) highlight the worsening economic situation and the intensification of other barriers (legal, information-related, or discriminatory), as has also occurred in previous epidemics [19]. All this points to situations in which the ability of these groups to access the health services and receive care may have diminished yet further.

In addition, new barriers may have been created as a result of adopting alternatives to face-to-face visits (online consultation, telephone, video call, telemedicine, etc.) and changes in attitudes and beliefs developed as a result of the pandemic. In this regard, the use of online consultation has grown as a way to mitigate difficulties in access [4], but not in the same way across all contexts [3,124]. Several articles included in the review described inequalities in access, reporting access problems related to digital literacy (lack of understanding of digital devices) or a lack of material resources (Internet connection, mobile devices) [49,85,89], a point also highlighted in other publications [125,126], alongside the perception that the care received in virtual consultation is impersonal [88]. These results are in keeping with numerous publications that have focused on the changes from face-to-face consultations to remote care and user satisfaction with the latter [29,80,127,128,129,130,131,132,133,134,135,136]. However, most studies are centered mainly on high-income countries, thus, further evidence in different contexts is required on the impact of remote care on access to the health services.

Likewise, one of the individual characteristics that has been most influential in terms of new barriers created by the pandemic is fear of contagion [83,84,85,86,88,89,94,95,97], an aspect that has been discussed in many publications, including opinion articles [113,137,138,139,140,141,142], and also played a highly significant role in previous epidemics as a factor causing problems or delays in seeking medical care [2,21,24,26,72,76,101,108,143,144]. Other factors reported include the stigma associated with seeking care [84,89,93,94,97], also described in other publications [109,145], and users playing down the need for medical treatment [84,94,98] and perceiving a lack of response on the part of the health services [85,88].

While it is true that various studies have identified both new barriers and the exacerbation of existing ones, the behavior of individuals in this type of public health emergency requires more in-depth analysis in order to steer the design of interventions to help counter these barriers, such as public health information campaigns or specific measures for vulnerable populations.

On a final note, this review has several limitations. In the first place the nature of the studies covered varies greatly, in terms of methodology (ways of measuring use of services, information sources, sample size, etc.), and of geographical areas and health systems analyzed; thus, it is not possible to draw comparisons between them. Secondly, articles that analyzed access to services due to COVID-19 in addition to other illnesses were excluded from the study. This decision was made in order to rule out bias towards activity and resources destined to the treatment of other diseases. Third, no studies focusing on the impact of the pandemic control measures on access to health services were found, probably due to the limited terms used in our search to capture this area. Moreover, it is also possibly due to the difficulties involved in distinguishing the impact of the measures from other effects of the pandemic. Finally, some articles may not have been considered on being published in other languages (Chinese, Arabic, etc.), so this analysis may have excluded relevant information and failed to consider certain contexts. In spite of these limitations, this is the first study to address changes in access from a global perspective, with a view to shedding light on gaps in knowledge that will require further research in the future.

## 5. Conclusions

This review analyzed studies that reported changes in health services’ utilization, and the factors that influenced the use of services for non-COVID-19 conditions, during the COVID-19 pandemic. Results vary according to the context analyzed, although, in general terms, they reflect the same trend, describing a general reduction in the use of health services, the exacerbation of preexisting barriers, and the emergence of new ones. This scoping review has shown that most studies conducted to date are limited in terms of scope and methodology and are centered mainly on the impact on specific conditions or population groups during the early stages of the pandemic, focusing mostly on secondary care. Furthermore, a significant gap in knowledge was detected on whether the services have recovered to pre-pandemic levels of care use and, if not, in which areas and for what reasons. Future studies should go into greater depth on the pandemic-related changes that have influenced access to health services (e.g., fear and socioeconomic difficulties), according to context and over the course of the different stages of the pandemic. In any case, as an ongoing phenomenon, the real impact of the COVID-19 pandemic is yet to be determined.

## Figures and Tables

**Figure 1 ijerph-19-01749-f001:**
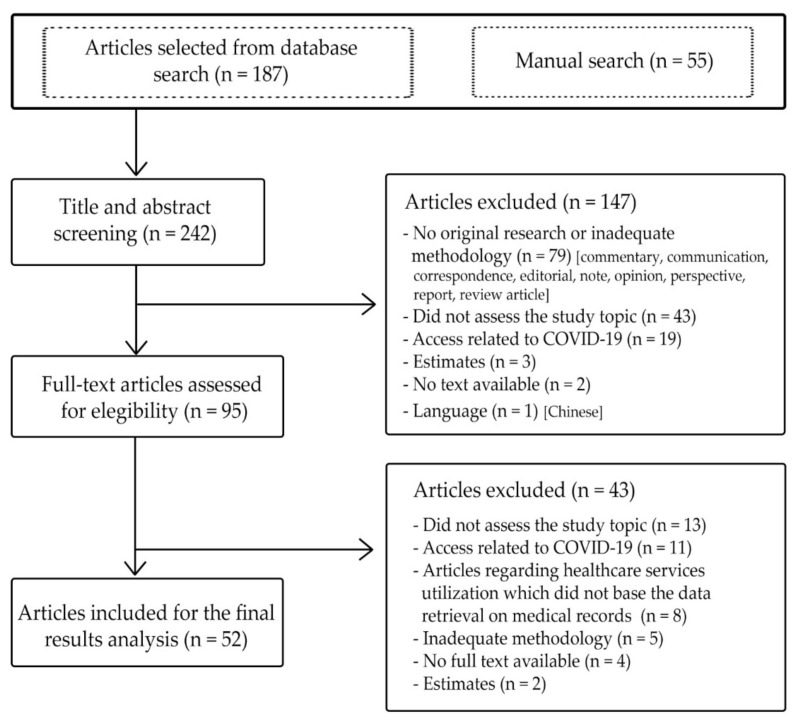
Flow chart of study selection process.

**Table 1 ijerph-19-01749-t001:** Changes in the utilization of health services and influencing factors during the COVID-19 pandemic in 2020.

First Author, Year	Data Source	Study Area	Study Population and Sample	Study Period	Health Service	Main Results
Abebe, 2021	Medical records from Tikur Anbessa Specialized Hospital	Ethiopia	Follow-up visits (*n* = 7717) and admissions (*n* = 3310) between December 2018–June 2019 and follow-up visits (*n* = 4597) and admissions (*n* = 2383) between December 2019–June 2020	December–June 2018–2019 vs. 2019–2020	General ^1^	Reduction in follow-up visits (40%) and admissions (28%) from March 2020, compared with the same period in 2019. Visits reduced especially among patients receiving renal, neurological, cardiac, and antiretroviral treatment (68–51.4%). No significant changes were observed among pediatric and adult admissions.
Howarth, 2021	Private health insurer claims records	United Kingdom (UK)	Claims to private health centers in the United Kingdom (*n* = aggregated data) ^2^	January 2018–August 2020	General	Reduction in healthcare claims in general (70%) from March 2020 (lockdown), undergoing an increase over the following months without reaching the pre-pandemic levels. Visits in mental health differed from the general pattern, with increased utilization (20%) compared to previous years. ^3^
Siedner, 2020	Africa Health Research Institute Demographic Health Surveillance System	KwaZulu Natal, South Africa	Visit to rural clinics (*n* = 46,523) ^2^	January–March vs. March–April vs. May–June 2018 vs. 2019 vs. 2020	General	Reduction in the number of childcare visits (50%), including preventive procedures, and sustained utilization of HIV services and adult outpatient clinics during the national lockdown (March–June 2020), compared to the previous periods. No significant changes were observed at a general level in the use of services. Childcare visits recovered over the following three months to pre-lockdown levels.
Whaley, 2020	Aggregate data on health insurance claims	United States of America (USA)	Population with health insurance in 2018 (*n* = 5.6 million), 2019 (*n* = 6.4 million), and 2020 (*n* = 6.8 million)	January–February and March–April 2018 vs. 2019 vs. 2020	General	Reduction in the utilization of a number of preventive services, elective procedures, and in-person office visits (different values according to the type of service or procedure) in March and April 2020. No significant changes in emergency care, maternal health, or medication prescription were observed. Utilization of telemedicine visits increased. Associated factors (AF): patients living in poor areas and most ethnic/racialized minorities were less likely to experience a reduction in in-person visits but also to have fewer visits of telemedicine.
Zhang, 2020	Aggregate data on China UnionPay Healthcare bank transactions	China	Health care transactions (*n* = 300 million) ^2^	January–March 2019 vs. December 2019–February 2020 vs. November 2019–April 2020	General	Reduction in daily expenditure on health (37.8%) and in number of visits to health services (40.8%) from January 2020. AF: higher probability of utilization of health services was associated with cities with lower rates of COVID-19 cases, less strict measures, and not located in the western region of the country.
Ojetti, 2020	Medical records from an urban tertiary teaching hospital	Italy	Admissions to the emergency department (ED) (*n* = 16,281)	February–March 2019 vs. 2020	ED	Reduction in ED admissions (37.6%) for several diseases in 2020 compared to 2019. There was an increase in triage emergency levels for ED admissions and in hospitalization rates (different values according to the type of admission).
Mahmassani, 2021	Administrative records of the emergency department of the American University of Beirut Medical Center	Beirut, Lebanon	ED visits, between November 2019–February 2020 (*n* = 16,271) and February–May 2020 (*n* = 8587)	November 2019–February 2020 vs. February–May 2020	ED	Reduction in general (47.2%) and pediatric (66.6%) ED visits from February 2020, compared to previous months.AF: higher probability of utilization was associated with elderly patients and those who required hospital admission and/or critical care, with a higher mortality rate, and with non-communicable diseases and bacterial infections.
Cano-Valderrama, 2020	Medical records from 3 hospitals	Spain	Patients who underwent emergency surgery in 2019 (*n* = 285) and 2020 (*n* = 117)	May–April 2019 vs. 2020	ED (Acute Surgery Care)	Reduction in surgeries (58.9%) during lockdown (March–May 2020). Longer waiting time between the onset of symptoms and arrival at the emergency room and a greater number of complications (especially in elective procedures) were observed.
Sokolski, 2021	Medical records from cardiology departments of 15 health centers in 12 countries	15 centers in 12 European countries and USA	Patients admitted to the emergency and cardiology departments (*n* = 54,331) ^2^	March–April 2019 vs. 2020	ED (cardiology)	Reduction in patient admissions (IRR 0.68) in 2020, compared to 2019, across the various pathologies treated (different levels of reduction that vary from IRR 0.66–0.68).
Tsioufis, 2020	Medical records from a tertiary General Hospital	Athens, Greece	Visits to the Emergency Cardiology Department and admissions to Cardiology Wards and Intensive Care Unit (*n* = aggregated data) ^2^	January–April 2018 vs. 2019 vs. 2020	ED (cardiology)	Reduction in visits to the emergency cardiology department during March (41.1%) and April (32.7%) 2020, compared to previous periods.
Ball, 2020	Aggregate data on hospital activity from 9 NHS hospitals	UK	Admissions and visits to ED for cardiovascular disease October 2018–May 2019: admissions (*n* = 599,372) and ED visits (*n* = 506,516); October 2019–May 2020: admissions (*n* = 513,703) and ED visits (*n* = 435,653)	October–May 2018–2019 vs. 2019–2020	ED (cardiovascular diseases)	Reduction in admissions (57.9%) and ED visits (52.9%) from March 2020, compared to the previous period.
Choi, 2021	Medical records from 6 hospitals	South Korea	Patients under 18 years of age seen in pediatric ED (*n* = aggregated data) ^2^	January 2017–November 2020	ED (pediatrics)	Reduction in pediatric ED visits (43.6%) in 2020 compared to previous years, although a significantly increased proportion of visits for injuries (9.4%) during the COVID-19 outbreak. AF: higher probability of use was associated with male patients.
Dopfer, 2020	Medical records from the University Hospital of Hannover	Hanover, Germany	Pediatric ED visits (*n* = 5424) ^2^	January–April 2019 vs. 2020	ED (pediatrics)	Reduction in pediatric ED visits (63.8%) from lockdown in 2020. AF: higher probability of using services was associated with patients under one year of age and cases requiring hospitalization, although not with intensive care admissions.
Finkelstein, 2021	Medical records from the Pediatric Emergency Research Network	Canada	Patients under 18 years of age who attended the ED in 2018 (*n* = 211,085), 2019 (*n* = 207,673), and 2020 (*n* = 159,049)	January 2018–January 2020 vs. January–March 2020 vs. March–April 2020	ED (pediatrics)	Reduction in weekly pediatric ED visits (58%), in re-visits (55%), in visits to trauma (increase in proportion of total visits), and to mental health (56 to 60% depending on the age group) from March 2020, compared to previous years. Increase in the proportion of ward (OR 1.39) and ICU (OR 1.2) admissions.
Goldman, 2020	Medical records from 18 pediatric emergency departments	British Columbia, Canada	Pediatric ED visits (0–16 years): March–April 2019 (*n* = 22,654); December 2019–January 2020 (*n* = 31,525); January–March 2020 (*n* = 26,654); March–April 2020 (*n* = 7535)	March–April 2019 vs. December 2019–January 2020 vs. January–March 2020 vs. March–April 2020	ED (pediatrics)	Reduction in visits to pediatric emergencies (57 to 70%), especially during the peak of the pandemic (March–April 2020), compared to previous periods. Admission proportion almost doubled (4% pre-pandemic to 7% during the peak pandemic period). Average acuity of illness was higher during the pandemic period.
Percul, 2021	Medical records from the Italian Hospital of Buenos Aires	Buenos Aires, Argentina	Patients under 18 years of age treated for appendicitis in 2019 (*n* = 117) and 2020 (*n* = 50)	March–August 2019 vs. 2020	ED (pediatrics)	Reduction in appendicitis admissions (25%) in 2020 compared to 2019, with no significant differences in the mean time to consultation. An increase in peritonitis cases was observed, although the incidence of complications decreased (not significant in both cases).
Yamamoto, 2021	Medical records from the Tokyo Metropolitan Children’s Medical Center	Tokyo, Japan	Patients under 18 years of age seen in pediatric ED between January–September 2017–2019 (mean *n* = 26,948 *) and January–September 2020 (*n* = 15,998)	January–September 2017–2019 vs. 2020	ED (pediatrics)	Reduction in pediatric ED visits (40.6%) in 2020 compared to previous periods, with an increase in the proportion of visits for exogenous causes (6.6% vs 3%). Visits increased slightly as of May until September2020, without reaching levels of previous years. ^3^
Kute, 2021	Medical records from the Kidney Disease Institute and Research Center	India	Patients treated in kidney disease services in 2019 (*n* = 109,572) and 2020 (*n* = 87,714)	January 2019–December 2020	SC: nephrology	Reduction in visits and admissions, transplants, and other elective procedures (different values according to the type of service or procedure) in 2020, compared to 2019. Slight increase in activity between July and October, without reaching previous levels, with a further reduction starting in November 2020. ^3^
Morris, 2021	NHS population-based datasets	UK	Patients referred for suspected or diagnosed colorectal cancer (*n* = monthly average mean) ^2^	January–December 2019 vs. January–October 2020	SC: oncology	Reduction in the monthly number of referrals for suspected cancer (63%) and for treatment (22%), colonoscopies (92%), and surgeries (31%) from April 2020, compared to 2019 and the preceding months. Relative increase in radiotherapy use (44%) due to increased use of short-course regimens. Monthly rate of referrals and other procedures returned to 2019 levels by October 2020.
Pareek, 2021	Medical records from the Gujarat Cancer Research Institute	Gujarat, India	Cancer patients visits to the oncology department between January–March (*n* = 4363) and March–May (*n* = 895) 2020	January–March vs. March–May 2020	SC: oncology	Reduction in visits from lockdown (different values according to the type of cancer) in March 2020, compared to the previous months. ^3^
Shi, 2021	Medical records from 13 pediatric tertiary cardiac centers	China	Patients who underwent cardiac surgery in 2018 (*n* = 19,398), 2019 (*n* = 19,620) and 2020 (*n* = 4740)	January–April 2018 vs. 2019 vs. 2020	SC: pediatric surgery	Reduction in the total surgical volume median (25 cases) compared to 2018 (148 cases) and 2019 (158 cases). Increase in the proportion of emergency operations (6.3%) during 2020, compared to previous years. Increase in patients followed-up via the internet or phone (26.4% in 2020 vs 9.5% and 8.9% in 2019 and 2018).
Ambrosetti, 2021	Medical records from the University Hospital of Geneva	Geneva, Switzerland	Admissions to the psychiatric ED from April to May 2016 (*n* = 702) and 2020 (*n* = 579)	April–May 2016 vs. 2020	SC: psychiatry	Reduction in admissions (17.5%) in psychiatric ED in 2020 compared to 2016. AF: the probability to be admitted was more associated with severe psychopathologies and single patients, who arrived by ambulance, with suicidal behaviors, behavioral disorders, and psychomotor agitation, and were more likely to be involuntarily hospitalized after consultation in ED.
Aragona, 2020	Medical records from the National Institute for Health, Migration and Poverty	Italy	Patients in a vulnerable situation who received at least one psychiatric intervention from February (*n* = 286) or March (*n* = 269) from 2017 to 2020 ^2^	February–March 2017 vs. 2018. vs. 2019 vs. 2020	SC: psychiatry	Reduction in visits (46.6%) to mental health in March 2020. Follow-up visits of patients from February to March decreased more (17.5% patients), compared to previous years (30% patients).
Jesenšek, 2021	Medical records from the Institute of Physical Medicine and Rehabilitation	Slovenia	Patients referred to rehabilitation in 2019 (*n* = 4132) and 2020 (*n* = 2317)	March–August 2019 vs. 2020	SC: rehabilitation	Reduction in the global volume of patients (44%), first visits (42%), and follow-ups (60.9%), as well as number of sessions (71.1%), from lockdown in March 2020, compared to 2019.
Farrugia, 2021	Medical records from Mater Dei Hospital	Malta	Admissions for acute exacerbations of chronic obstructive pulmonary disease in 2019 (*n* = 259) and 2020 (*n* = 119)	March–May 2019 vs. 2020	SC: respiratory diseases	Reduction in admissions (54.2%) in 2020 compared to 2019. Increase in the mortality of admitted patients (19.3% vs. 8.4%).
Burt, 2021	Medical records from Kawempe National Referral Hospital	Kawempe, Uganda	Visits to antenatal (*n* = 14,401), maternal health (*n* = 33,499), childcare (*n* = 111,658) and SRH (*n* = 57,174) services ^2^	July 2019–December 2020	SC: sexual and reproductive health (SRH) (antenatal, maternal, pediatrics, and family planning)	Reduction in antenatal, childcare, and family planning visits, as well as hospital deliveries (different values according to the type of service or procedure), during the lockdown months (March–June 2020), compared to previous months, without clear subsequent recovery. Increase in pregnancy complications and fetal and infant outcomes.
Das Neves, 2021	Medical records from Marrere Health Center and monthly official statistics from the Ministry of Health	Nampula, Mozambique	Visits to SRH services (*n* = aggregated data) ^2^	March–May 2019 and 2020	SC: SRH (maternal and child health)	Reduction in family planning visits (28%), elective C-sections (28%), first antenatal visits (26%), hospital deliveries (4%) (increase in out-of-hospital deliveries by 74%), and child vaccination (20%). Only hospital deliveries drops were statistically significant.
Jensen and McKerrow, 2020	Medical records from the KwaZulu-Natal District Health Information System	KwaZulu-Natal district, South Africa	Visits to child health services (aggregated data) ^2^	January 2018–June 2020	SC: SRH (maternal and child health)	Reduction in clinical visits (36%), hospital admissions (50%), delivery of services (from 6% to 54% depending on the service) in children under 5 years of age from March 2020. Modest increase in clinic visits as of May 2020, without reaching levels of preceding years. Among delivery of services, immunization coverage increased almost to pre-pandemic levels.
Justman, 2020	Medical records from a tertiary referral center	Haifa, Israel	Pregnant women (*n* = aggregated data) ^2^	March–April 2019–2020	SC: SRH (maternal and child health)	Reduction in visits (from 18.1% to 36.4% according to the type of visit), deliveries (17.1%) and admissions (22.3%) to the obstetrics and gynecology department in 2020 compared to 2019. No significant changes were observed in the rate of C-sections, although a greater number of vaginal births during the outbreak (16.7% in 2020 vs. 6.8% in 2019), between the two periods.
KC, 2020	Data collected from a prospective observational study in 9 hospitals (SUSTAIN and REFINE studies)	Nepal	Pregnant women (*n* = 21,763)	January–March vs. March–May 2020	SC: SRH (maternal and child health)	Reduction in hospital deliveries (52.4%), especially vaginal births, from lockdown in March 2020. Increase in preterm births (24.5% before lockdown vs. 26.2% during lockdown), neonatal deaths (13 per 1000 livebirths vs. 40 per 1000 livebirths) and women admitted with complications during labor (6.7% vs. 8.7%, not statistically significant). AF: lower utilization of SRH services was less likely among users of disadvantaged ethnic groups and poor perceived quality of care.
Marqués, 2021	Medical records from the Cambridge University Hospitals NHS Foundation Trust	Cambridge, UK	Women complaining of a 1st episode of reduced fetal movements in 2019 (*n* = 810) and 2020 (*n* = 803)	March–April 2019 vs. 2020	SC: SRH (maternal and child health)	Reduction in 1st visits for reduced fetal movements (RFM) during 2020, compared to the same period in 2019 (18% vs. 22%). AF: primiparous women were more likely to attend with RFM.
Shakespeare, 2021	Medical records from Mpilo Central Hospital	Zimbabwe	Women who gave birth from January to June 2020 (*n* = aggregated data) ^2^	January–March vs. April–June 2020	SC: SRH (maternal and child health)	Reduction in visits (5.8%) for hospital deliveries from April 2020, compared to previous months. No significant changes were observed in maternal or perinatal mortality and morbidity, nor in workload, although the number of deliveries and C-sections fell. Neonatal deaths increased, not significantly.
Spurlin, 2020	Medical records from the New York Presbyterian—Columbia University Irving Medical Center	New York, USA	Patients who attended OB-GYN (obstetrics–gynecology) services from February to March 2020 for emergency visits (*n* = 275), GYN surgeries (*n* = 212), OB surgeries (*n* = 237), and from March to April 2020 for emergency visits (*n* = 79), GYN surgeries (*n* = 79), OB surgeries (*n* = 181)	February–March vs. March–April 2020	SC: SRH (obstetrics and gynecology)	Reduction in the average weekly OB-GYN ED consults (60.3%) and GYN surgeries (79.3%), whereas OB surgeries remained stable, from March 2020 compared to the previous period. No significant differences in the proportion of OB-GYN ED consults and GYN surgeries were observed, although the proportion of OB surgeries increased significantly (54.6% before March vs. 79.7% from March 2020).
Chiba, 2021	Medical records from the Medical Center of the University of Southern California and Los Angeles County	Los Angeles, USA	Patients admitted to trauma in 2019 (*n* = 1143) and 2020 (*n* = 1202)	March–June 2019 vs. 2020	SC: traumatology	Increase in the number admissions (different values according to the type of trauma) during the analyzed period of 2020, compared to 2019. Increase in admissions due to falls (32.4%) (especially elderly), injuries from the use of weapons (39.3%), suicides (38.5%, not statistically significant), and positivity in the use of substances (52.1% in 2020 vs. 40.2% in 2019). Reduction in severe trauma (38.7% vs. 46.7%), mortality (4.1% vs. 5.9%), and ICU admission rates (26.3% vs. 31.5%). There were non-significant reductions in admissions due to traffic accidents (pedestrian or motor).
Horan, 2021	Medical records from the National Neurosurgical Center at Beaumont Hospital	Dublin, Ireland	Referrals to the trauma department in 2019 (*n* = 527) and 2020 (*n* = 437)	March–May 2019 vs. 2020	SC: traumatology	Reduction in trauma referrals (17.1%) in 2020 compared to 2019. No significant changes were observed between the profiles most associated with shunts between the two years, although there were changes in the type of diagnosis (fewer brain and spinal injuries and cranial fractures). AF: referrals were more likely among men, people over 60, alcohol consumers.
Alexander, 2020	IQVIA National Disease and Therapeutic Index	USA	Visits to primary care (*n* = 875.6 million) ^2^	January 2018–June 2020	PC	Reduction in PC health services (21.4%) in 2020 compared to 2018 and 2019. Decreases in in-person visits (50.2%) and increases in telemedicine visits (1.1%) were observed. Evaluations and medication prescriptions were less frequent.
Sato, 2021	Administrative claims from the DeSC database (health insurance claims)	Japan	Patients with chronic neurological diseases ^2^	March–November 2020	PC	Reduction in visits for different chronic neurological diseases (RR 0.9), except one that increased (migraines, RR 1.15), from April 2020. Telephone appointments were most frequently used in April–May (representing 5% of the visits), especially in the case of migraines (OR 2.08). The changes yielded different effects depending on the disease.
Song, 2021	Medical records from the Independence Blue Cross	USA	Women who had mammograms from January 2018 to March 2020 for screening (*n* = 213,168) and diagnosis (*n* = 55,879), and from March to July 2020 for screening (*n* = 27,970) and diagnosis (*n* = 10,233)	January 2018–March 2020 vs. March–July 2020	Preventive services	Reduction in the volume of screening (58%) and diagnostic (38%) mammograms from March 2020, compared to the preceding months and to the previous years. Increase in activity from May 2020, remaining 14% below previous months levels. **AF**: greater use was associated with women with a previous diagnosis.

^1^: General health services include different levels of care or type of service; ^2^: aggregated data and/or no specification of the different periods of analysis; ^3^: studies that did not analyze whether the changes were statistically significant (the rest of articles presented results statistically significant); AF: associated factors; ED: emergency department; ICU: intensive care unit; IRR: incidence rate ratio; OB-GYN: obstetrics–gynecology; OR: odds ratio; PC: primary care; RR: relative risk; SC: secondary care; SRH: sexual and reproductive health.

**Table 2 ijerph-19-01749-t002:** Quantitative studies on potential access related to the characteristics of the services and the population during the COVID-19 pandemic in 2020.

First Author, Year	Data Collection Method	Study Area	Study Population and Sample	Study Period	Health Service	Main Results
Kahraman et al., 2021	Online survey	Turkey	Patients with lysosomal storage disease in enzyme replacement therapy (*n* = 75)	July–October 2020	SC: endocrinology	Characteristics of the services: lack of resources (hospital beds) Characteristics of the population: fear of contagion, difficulties in obtaining medication, transport difficulties
Nicholson et al., 2020	Online survey	Ireland	Parents of children under 16 (*n* = 1044)	June 2020	SC: pediatrics	Characteristics of the population: fear of contagion, perception of overuse of services or lack of need, fear of being judged for seeking care, poor understanding of government messages, concern regarding travel (avoiding public transport).
Benjamen et al., 2021	Online survey (*n* = 77, of which 11 were interviewed in depth)	Ottawa, Canada	Doctors with experience caring for refugee populations (*n* = 77)	May–August 2020	SC: psychiatry	Characteristics of the services: limited availability of providers and community resources, slight increase in the offer of virtual care psychotherapy. Characteristics of the population: fear of contagion, perceived lack of services, technological barriers
Halley et al., 2021	Online surve	USA	Relatives (*n* = 139) and patients affected by undiagnosed rare diseases (*n* = 275)	April–June 2020	SC: undiagnosed rare diseases care	Characteristics of the services: barriers to access essential services (difficulties in contacting services, procedures re-scheduled, lack of medical supplies, insufficient telemedicine care offered), restrictions on companions. Characteristics of the population: fear of COVID-19 contagion; impact on physical and mental health (stress due to not being able to receive treatment or as an aggravating factor of the disease).
Adelekan et al., 2021	Semi-structured interviewer-administered questionnaire	Nigeria	Head nurses and midwives in primary health centers (*n* = 307)	March–September 2020	SC: SRH (maternal and child health)	Characteristics of the services: difficulties regarding out-of-stock drugs and contraceptives. Characteristics of the population: economic difficulties (not being able to afford cost of transportation).
Karavadra et al., 2020	Online survey	UK	Women who were pregnant or gave birth during the COVID-19 pandemic (*n* = 1451)	May 2020	SC: SRH (maternal and child health)	Characteristics of the services: reduced frequency of scans, redistribution of services in different “zoned areas” based on “COVID wards” and “non-COVID” wards, lack of information, ban on presence of partner. Characteristics of the population: fear of contagion, perception of “impersonal care” from virtual consultations.
Khan et al., 2021	Online survey	64 middle- and low-income countries of Africa, Asia, and Latin America	Health professionals from tuberculosis treatment (*n* = 567) and HIV (*n* = 346) services	May–August 2020	SC: tuberculosis and HIV care	Characteristics of the services: lack of material and medical supplies, difficulties in obtaining medical treatment, lack of alternatives for non-face-to-face care (e.g., telemedicine), postponement of visits for diagnoses and treatments. Characteristics of the population: fear of contagion, stigma, difficulties in accessing health services (alterations in transportation, restrictions), worsening economic situation.
García-Rojo et al., 2021	Medical records (Hospital 12 de Octubre)	Spain	Patients on the waiting list for urological surgery (*n* = 350)	May 2020	SC: urology	Characteristics of the services: increased waiting times for urological surgeries (designated as elective).

SC: secondary care; SRH: sexual and reproductive health.

**Table 3 ijerph-19-01749-t003:** Qualitative studies on potential access related to the characteristics of the services and the population during the COVID-19 pandemic in 2020.

First Author, Year	Data Collection Method	Study Area	Study Population and Sample	Study Period	Health Service	Main Results
Zambrano et al., 2021	Online semi-structured interviews and life histories	Colombia and Peru	Venezuelan migrant populations living in large cities in Colombia (*n* = 96) and Peru (*n* = 34)	July–September 2020	General	Characteristics of the services: access to health services linked to legal immigration status. Characteristics of the population: severe economic difficulties, perceived discrimination of healthcare services on the basis of nationality
Das Neves et al., 2021	Semi-structured interviews by phone (12) and in-person (9)	Nampula, Mozambique	Health professionals (*n* = 9), traditional birth attendants (*n* = 6) and patients (*n* = 6)	March–May 2019 and 2020	SC: SRH (maternal and child health)	Characteristics of the services: limited resources (workforce), increase in waiting times Characteristics of the population: fear of contagion, avoiding healthcare except in emergencies
Gichuna et al., 2020	Semi-structured interviews via mobile phone (phone call or videocall)	Kariobangi, Roysambu, and Jogoo Road areas (Nairobi, Kenya)	Sex workers from the study areas (*n* = 117), and health professionals from the Bar Hostess Empowerment and Support Program Centers (*n* = 15)	April–May 2020	SC: SRH (family planning) and HIV care	Characteristics of the services: reduced activity, lack of medical supplies Characteristics of the population: economic difficulties, stigma
Hailemariam et al., 2021	Online focal groups (6) and semi-structured interviews (9)	Kebeles, Ethiopia	Pregnant women who did not attend SHR services and health workers	September–November 2020	SC: SRH (maternal and child health)	Characteristics of the population: perceived low quality of services, fear of contagion, stigma, playing down care needs, refusal to attend antenatal services.
Mizrak Sahin, and Nur Kabakci, 2020	Semi-structured interviews by phone	Turkey	Pregnant women	During 2020, months not specified	SC: SRH (maternal and child health)	Characteristics of the services: elective visits were cancelled or postponed, difficulties in getting first visits. Characteristics of the population: fear of contagion, difficulties in contacting services for first visits.
Dos Santos et al., 2021	7 semi-structured interviews by phone	Ribeirão Preto, Brazil	Patients over 18 years old undergoing treatment for tuberculosis	June–August 2020	SC: tuberculosis care	Characteristics of the population: economic difficulties, fear of contagion
Ahmed et al., 2020	Workshops and in-person meetings in three pre-pandemic phases (semi-structured interviews, group and individual meetings), and a fourth phase via mobile phone	Bangladesh, Kenya, Nigeria, Pakistan	Health professionals (medical doctors, nurses, community health workers and assistants, pharmacists, and patent medicine vendors), pregnant women and women with children, health service managers	March 2018–May 2020	PC	Characteristics of the services: worse access to services that were difficult to access before the pandemic (mental health, gender-based violence services), and preventive services, increase in cost of healthcare, lack of drugs and medical supplies. Characteristics of the population: fear of contagion, economic difficulties.
Danhieux et al., 2020	Online semi-structured interviews	Belgium	General practitioners, nurses, and dieticians (*n* = 21) in primary care who work individually, monodisciplinary or in multidisciplinary groups	April–June 2020	PC	Characteristics of the services: limited resources to treat, identify, and contact non-COVID-19 patients because of the redistribution of resources, especially among high-risk and vulnerable patients.

PC: primary care; SC: secondary care; SRH: sexual and reproductive health.

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
