# Peer review of "Changes in Access to Health Services during the COVID-19 Pandemic: A Scoping Review"

_ijerph, 2022, doi:10.3390/ijerph19031749_

Round 1

Reviewer 1 Report

This manuscript presents a scoping review of articles presenting access to health care for non-COVID-related pathologies during the COVID-19 pandemic. The authors used the framework of Aday and Anderson to classify and present their findings. Thank you for putting your significant efforts to study this question. The work is very interesting and needs refinement.

To be able to better understand the value of the approach presented in this manuscript, there is a need for additional clarifications in the background, methods and other sections. For example:

  1. Lines 30-33: The provided references do not sufficiently clarify the statement on these lines.
    • What is meant by the "context of the pandemic"?
    • What is meant by "the diverse strategies that were adopted to tackle it (i.e., "context of the pandemic"?)?
    • "An impact on the health of the population" is described as "above and beyond that caused by the COVID/19 itself". You need to clarify your study hypothesis, otherwise, the study premises is insufficient.
  2. Lines 37-38:
    • Who's is the classification of services, that you mention?
    • Do you imply that classification is inappropriate?
    • If so, why? Provide referenced arguments for this statement.
    • How does that statement (or assumption) relate to your study aim?
  3. Lines 39-44: This is a too-long sentence, and unclear. What is that you want to say?
  4.  Lines 44-48: Need to be concrete.
    • "This" is not clear, as it refers to the previous too-long sentence. Need to be concrete.
    • "may have affected vulnerable population".  How? Who in particular?
    • "axes of inequality" What are you implying?
    • "The adoption of these measures" is the problem? Please be prepared to show evidence of the situation before their adoption.
    • "may have led to other repercussions", is too vague, I do not understand.
    • where is the evidence of a "greater burden of disease"?
    • where is the evidence of "worsening inequalities in access to health care"?
  5.  Lines 49-56: This is an unclear sentence.
    • Whose experience?
    • What kind of experience?
    • What kind of "short and long term" "problems related to other diseases" has been generated by epidemics and outbreaks?
    • Where is the scientific argument for such a declaration?
  6. Line 57-66: What kind of impact on the access you were looking for? Are you sure you have good insight for making a firm comparison such as this? Who has found a few literature reviews? Please, rephrase, so you do not imply that decision-makers were not responsible and not monitoring their actions,  so do not know the impact on the access to services they provide.
  7. Line 70: Please, provide the relevant reference for "the most frequently used models for the analysis of access to health services". Please, explain the value of this model in comparison to other models you have for studying access to health care. Why this model is particularly relevant for studying your hypothesis and article aim?
  8. Line 80-83: The study aim is to "summarise the knowledge on the impact of
    •  the COVID-19 pandemic, and 
    •  Mitigation measures

in access to health services for non-COVID-related conditions.

So the question is: Where do you describe mitigation measures? How does the study hypothesis/aim relate to the study concept and Aday and Anderson model? 

Methods: As explained, the framework is used only to identify variables of interest and to group them.

Line 113: What does "population records" mean?

The results section only provides a list of variables grouped by the chosen model.

Discussion: Many parts of the discussion section should be moved to the background. Not all study limitations are described.

The conclusion is largely inappropriate. Why there is no synchronisation between the title, the methods and the first sentence in the conclusion section?

Reviewer 2 Report

This research aims to synthesize the changes in access to Health Services during the COVID-19 Pandemic in a scope review. In the selected 53 papers, they generalized reduction in use. It also indicates that the most frequent access barrier for non-COVID-19 conditions related to the services was a lack of resources, while barriers related to the population were predisposing and enabling characteristics. The research concluded that more studies are required on the subsequent phases of the pandemic, and recommend focusing on the factors that have influenced access and using a mixed-methods approach.

The research is of great significance by contributing to our knowledge of the changes that have arisen in the COVID-19 Pandemic, and their implications for equity of access and the resilience of national health systems across the world.

There are some points to improve.

  1. The results and findings are scattered. The abstract does not describe clearly what are the changes in access to Health Services during the COVID-19 Pandemic. Meanwhile, the conclusion should summarize the biggest findings of this research, rather than merely suggesting future research directions.
  2. There are a lot of research on this topic, but the research chose to conduct a scope review across different countries and various health systems; thus, this research needs to clarify the meaning of doing so. It is important to stress the benefits in the introduction part and in the discussion. The research may classify the selected studies into groups, and analyze the similarities and differences across studies.
  3. The research used search items such as “Health policy”, which does not seem to be closely related to its research topic.

Also, this paper used “healthcare” as an important search item, but the healthcare is a very broad definition, and many studies may not use this word directly. For example, some papers may use the phrases such as “mental health care”, “dental care”, “long term care”, or “hospital service”, they may not use “healthcare” at all. How did this research capture those?

  1. In general, the scope review could summarize and present the information better.
  2. There are some grammar errors and typos to correct, e.g., “the studie that analyzes changes”.

Reviewer 3 Report

Firstly, thank you for opportunity to review very interested article. I don't feel qualified to judge about the English language and style due to not native language.

  1. The title reflect the main subject about access to health services during the COVID-19, title was clear and easy to understand.
  2. The abstract summarize and reflect the work described in the manuscript.
  3. The key words reflect the focus of the manuscript.
  4. The manuscript adequately describe the background, present status, and significance of the study. The authors explain effect of COVID-19 with health services. I suggested the authors to identify "health service" meaning because effect from COVID-19 related in pre-hospital and in-hospital, the authors should clarify scope of study.
  5. The manuscript describe methods in adequate detail, study subjects were clear.
  6. The research objectives achieved by the experiments used in this study.
  7. The manuscript interpret the findings adequately and appropriately, highlighting the key points concisely, clearly, and logically.
  8. Tables and figures sufficient, good quality and appropriately illustrative of the paper contents.
  9. The manuscript cite appropriately the latest, important and authoritative references in the introduction and discussion sections.

Round 2

Reviewer 1 Report

The authors have adequately responded to the comments and the topic has been better explained. I kindly advise a few more issues that emerged in the revised paper to be resolved.

Abstract, line 63:  whose recovery? what is recovered?

Introduction: Lines 228-237 explain well the framework and this is a good addition. However, here needs to explain why this framework fits best to the study objective in comparison to other widely-used and capital models that are mentioned in the Notes to reviewer? 

Delete "and the mitigation measures adopted" from the article aim,  there were no adequate key terms for the search of "mitigation measures". That is why there were no sufficient results to support the claim on lines 400-401.

Methods

Delete text on lines 400-401: "No studies...of the pandemic".

Need to explain the approach used to categorise your study findings:  "slight", "heavy", etc.

Results

3.1.  and 3.2 Whole sections: Please, provide figures where possible and SMART information to support study findings such as "a slight recovery" (and a recovery of what)? "a heavy drop in use"? "fewer visits", "lower utilisation",  "higher incidence",  "a decrease in resources", sorry these are examples of incomplete comparisons. I assume you know what you want to say but it is not clear to the reader.

What exactly was measured and how, whit what indicator and when, where was measured? and then explain why you use "a slight"

Discussion: Authors need to say the limitation of

  •  the model
  • key terms used.

Also in discussion authors need to say simply and specifically the advantage of its model over the others available.

Delete text on lines 902-905: "Moreover,....of the pandemic" - there were no key terms for the search of "control measures".

Delete lines 917-918: "It is important....numerous actors". Access is not a phenomenon!

Correct text on lines 917- 922, and say: Our review showed a limited number of qualitative studies that focused on access in the searched period and type of service. And think and explain underlying logical reasons, including those that related to the limitations of your study methods and approach.

Thank you! Please add a few more efforts to increase the chances of quoting your paper, when published.

Reviewer 2 Report

The authors have responded well to my previous critiques.

Author Response

We would like to thank the reviewer again for their comments, which helped to improve our manuscript.